# Antidepressant-Like Effect and Mechanism of Action of Honokiol on the Mouse Lipopolysaccharide (LPS) Depression Model

**DOI:** 10.3390/molecules24112035

**Published:** 2019-05-28

**Authors:** Bo Zhang, Ping-Ping Wang, Kai-Li Hu, Li-Na Li, Xue Yu, Yi Lu, Hong-Sheng Chang

**Affiliations:** 1School of Chinese Materia Medica, Beijing University of Chinese Medicine, Beijing 102488, China; zxcvbo613@163.com (B.Z.); pingping.wang@bettapharma.com (P.-P.W.); hkl1024@163.com (K.-L.H.); 2School of Chinese Medicine, Beijing University of Chinese Medicine, Beijing 100029, China; 18201012962@163.com (L.-N.L.); loving0506@126.com (X.Y.); luyi@bucm.edu.cn (Y.L.)

**Keywords:** honokiol, antidepressant effect, anti-inflammatory, tryptophan–kynurenine pathway, IDO, NF-κB

## Abstract

There is growing evidence that neuroinflammation is closely linked to depression. Honokiol, a biologically active substance extracted from *Magnolia officinalis*, which is widely used in traditional Chinese medicine, has been shown to exert significant anti-inflammatory effects and improve depression-like behavior caused by inflammation. However, the specific mechanism of action of this activity is still unclear. In this study, the lipopolysaccharide (LPS) mouse model was used to study the effect of honokiol on depression-like behavior induced by LPS in mice and its potential mechanism. A single administration of LPS (1 mg/kg, intraperitoneal injection) increased the immobility time in the forced swimming test (FST) and tail suspension test (TST), without affecting autonomous activity. Pretreatment with honokiol (10 mg/kg, oral administration) for 11 consecutive days significantly improved the immobility time of depressed mice in the FST and TST experiments. Moreover, honokiol ameliorated LPS-induced NF-κB activation in the hippocampus and significantly reduced the levels of the pro-inflammatory cytokines; tumor necrosis factor α (TNF-α), interleukin 1β (IL-1β), and interferon γ (IFN-γ). In addition, honokiol inhibited LPS-induced indoleamine 2,3-dioxygenase (IDO) activation and quinolinic acid (a toxic product) increase and reduced the level of free calcium in brain tissue, thereby inhibiting calcium overload. In summary, our results indicate that the anti-depressant-like effects of honokiol are mediated by its anti-inflammatory effects. Honokiol may inhibit the LPS-induced neuroinflammatory response through the NF-κB signaling pathway, reducing the levels of related pro-inflammatory cytokines, and furthermore, this may affect tryptophan metabolism and increase neuroprotective metabolites.

## 1. Introduction

Depression is a mental illness characterized by low mood. The clinical manifestations include low energy, slow thinking, reduced speech and movement, loss of interest or pleasure, and, in severe cases, suicidal tendencies [1]. Depression is ranked second in the WHO ‘global burden of disease’ rankings [2], just behind cancer. At present, depression is treated mainly by the use of drugs. The ‘monoamine hypothesis’, the decrease of monoamine neurotransmitter levels in the synaptic cleft, is still recognized as the major pathogenesis of depression in clinical settings [3]. Most classical antidepressants, such as tricyclic antidepressants (TCAS), tetracyclic antidepressants (HCA), monoamine oxidase inhibitors (MAOI), and selective serotonin (5-HT) reuptake inhibitors (SSRIs), all presently on the market, are based on this theory. However, current monoamine neurotransmitter-based drugs are ineffective in about 30% of patients with depression, and long-term use can cause side effects such as cardiotoxicity, sleep disorders, and sexual dysfunction [4,5]. Therefore, it is important to elucidate other possible mechanisms that may cause depression, and to develop drugs to inhibit these pathways with fewer adverse reactions and side effects.

Many studies have shown that the pathogenesis of depression is closely related to the inflammatory response [6,7,8]. Clinical studies have found that levels of inflammatory cytokines, such as tumor necrosis factor α (TNF-α) and interleukin 1β (IL-1β), are abnormally elevated in the peripheral blood and cerebrospinal fluid of patients with depression, and that inflammatory cytokines in patients with depression can return to normal levels after administration of antidepressants [9,10,11]. There have also been similar indications in related animal studies: An abnormal increase in inflammatory cytokine levels has been observed in acute and chronic stress models of depression [12,13,14]. Direct injection of inflammatory cytokines, such as TNF-α and IL-1β, into the brain of animals can produce depression-like behaviors, such as loss of pleasure, decreased activity, and decreased food intake [15,16,17]. In addition, other studies have shown that depression in the inflammatory state is often accompanied by a disorder of the tryptophan–kynurenine (TRP–KYN) metabolic pathway, while indoleamine 2,3-dioxygenase (IDO), an enzyme involved in tryptophan metabolism, is a key factor connecting inflammation and depression [18,19]. Clinical studies have found that patients with depression have a higher KYN/TRP ratio than non-depressed patients, suggesting that IDO activation is associated with depression [20]. These studies all indicate that neuroinflammation is involved in the development of depression, contributing to a new understanding of the illness.

Honokiol is a bioactive polyphenolic substance and is one of the main active components extracted from the plant, *Magnolia officinalis*, which is widely used in traditional Chinese medicine [21]. Modern studies have shown that honokiol has few toxic side effects and many pharmacological activities, including anti-inflammatory, anti-oxidant, free radical scavenging, antibacterial, anti-cancer, and nerve cell protecting properties [22,23,24,25,26,27,28]. Honokiol may exert its anti-inflammatory effects through multiple pathways, such as down-regulation of the MAPK and NF-κB signaling pathways, causing inhibition of lipopolysaccharide (LPS)-induced iNOS and COX-2 expression [29]; inhibition of the phosphatidylinositol 3-hydroxykinase/protein kinase pathway (PI3K/AKT) [30] and; inhibition of the activation of the NF-κB signaling pathway by the inhibition of IκB kinase activation [31]. Additionally, honokiol is a small molecule that can easily pass through the blood–brain barrier, providing direct contact with the nerve tissue, thus, it has a direct effect on nerve cells [32]. Studies have shown that honokiol can improve LPS-induced anxiety-like behavior by inhibiting the production of inflammatory cytokines [33].

Our previous study [34] found that honokiol improved depression-like behavior in acute and chronic stress mice. In the present study, we established a mouse depression model by peripheral administration of LPS, and measured the concentration of the pro-inflammatory cytokines, TNF-α, IL-1β, and interferon γ (IFN-γ), in peripheral blood, the concentration of IFN-γ in the brain, the expression of NF-κB P65 and IDO in the hippocampus, and the concentration of calcium in the brain. Measurement of TRP–KYN-related metabolites further confirmed the regulatory and neuroprotective effects of honokiol on depression-induced behavior in this model.

## 2. Results

### 2.1. The Effect of Honokiol on Autonomous Activity

The purpose of spontaneous activity tests is to exclude the interference of immobile time in the forced swimming test (FST) and tail suspension test (TST). There was no difference in autonomous activities, such as the distance travelled, speed of movement, and the frequency of residence in the central region and the marginal region, between the control, LPS, and LPS + honokiol groups (Figure 1). This indicates that honokiol has no obvious excitatory or inhibitory effects on the central nervous system, therefore, any false positive reaction in the behavioral despair tests can be excluded.

### 2.2. The Effects of Honokiol on Immobility Time in the Forced Swimming Test and Tail Suspension Test

In the FST and TST, the immobility time of the animals in the LPS group increased significantly compared with those of the control group (*p* < 0.001) (Figure 2A,B). These results indicate that we have successfully established a mouse model of LPS-induced depression. Compared with those of the LPS group, the FST and TST immobility times of the honokiol group were significantly reduced (*p* < 0.01) (Figure 2A,B). This indicates that honokiol can significantly improve the behavioral despair of mice with depression, suggesting that it has anti-depression-like effects.

### 2.3. The Effect of Honokiol on Pro-Inflammatory Cytokines

To investigate the anti-inflammatory effects of honokiol, the serum concentrations of TNF-α, IL-1β, and IFN-γ, were determined by radioimmunoassay. The levels of all three cytokines were significantly increased in the serum of the LPS group mice in comparison with those in the control group (TNF-α and IFN-γ: *p* < 0.001; IL-1β: *p* < 0.01). In contrast, the levels of the three cytokines were significantly lower in the honokiol group than those in the LPS group (TNF-α and IL-1β: *p* < 0.05; IFN-γ: *p* < 0.001) (Figure 3A–C).

IFN-γ is the main inducer of IDO activation. To evaluate whether any activation of IDO in the hippocampal area of the brain is accompanied by an increase of IFN-γ in the brain, an ELISA kit was used to detect IFN-γ in brain tissue. The content of IFN-γ in the brain tissue of the LPS group was significantly increased compared with that in the control group (*p* < 0.05); however, it was significantly lower in the brain tissue of the honokiol-treated animals than it was in the tissue of the LPS group (*p* < 0.01) (Figure 3D).

These results indicate that honokiol can significantly inhibit the release of pro-inflammatory cytokines induced by LPS and that it has significant anti-inflammatory effects.

### 2.4. The Effects of Honokiol on mRNA and Protein Expression of Hippocampal IDO and NF-κB

To investigate whether activation of NF-κB and IDO is involved in LPS-induced depression, the expression of IDO1 and NF-κB P65 mRNA in the hippocampus of the mice was analyzed by real-time quantitative PCR. The mRNA expression of both NF-κB P65 and IDO1 was significantly higher in the hippocampi of the LPS group mice compared with that in the control group (NF-κB P65: *p* < 0.001; IDO1: *p* < 0.05) (Figure 4A,B). After treatment with honokiol, the mRNA expression of both genes was significantly reduced (*p* < 0.001) (Figure 4A,B).

Western blotting was used to study the levels of IDO and NF-κB P65 protein expression in the hippocampi of the mice. The expression of IDO and NF-κB P65 proteins was significantly increased in the LPS group compared with that in the control group (*p* < 0.01) (Figure 4C–E). Compared with the LPS group, the honokiol group had significantly reduced the hippocampal expression of IDO and NF-κB P65 proteins (*p* < 0.01) (Figure 4C–E).

Both the western blot and the real-time quantitative PCR analyses indicate that honokiol can inhibit the activation of IDO and NF-κB induced by LPS.

### 2.5. The Effect of Honokiol on TRP-KYN Metabolites

TRP–KYN-related metabolites were detected by HPLC–MS. The levels of 5-HT and TRP in the serum of the animals in the LPS group were significantly lower than those in the serum of the control group (*p* < 0.01 and *p* < 0.05, respectively) (Figure 5A,B); however, the levels of KYN and quinolinic acid (QUIN) were significantly increased (*p* < 0.01 and *p* < 0.05, respectively) (Figure 5C,E). The levels of 5-HT were significantly increased, and those of KYN and QUIN were significantly reduced, compared with the LPS group, after treatment with honokiol (*p* < 0.05) (Figure 5A,C,E). The serum KYN/TRP and kynurenine/kynurenic acid (KYN/KYNA) ratios in the LPS group mice were significantly higher than those in the control group (*p* < 0.01) (Figure 5F,G). Compared with the LPS group, the serum KYN/TRP and KYN/KYNA ratios were significantly reduced by honokiol (*p* < 0.05). These results indicate that honokiol is involved in the regulation of TRP–KYN metabolism.

### 2.6. The Effect of Honokiol on the Concentration of Free Calcium in the Brain

The concentration of free calcium ions in mouse brain cells was observed by flow cytometry. Ca^2+^ ion fluorescence in the brain cells of the honokiol group was significantly lower than that of the LPS group (*p* < 0.01) (Figure 6). This demonstrates that honokiol can decrease the amount of free calcium in the brain cells of LPS model mice and inhibit calcium overload.

## 3. Discussion

Relevant studies have shown that intraperitoneal injection of LPS can lead to an acute inflammatory response, causing depression-like behavior and cognitive changes in mice [35]. LPS activates the innate immune response, resulting in the secretion of various pro-inflammatory cytokines, including TNF-α, IFN-γ, and IL-1β, and the induction of neuroendocrine and neurochemical changes, leading to depression [36,37,38].

Neuroinflammation is considered to be an important pathological cause of depression, and this corresponds to the cytokine hypothesis of the pathogenesis of depression [39]. Cytokines, activated and secreted by immune cells, are signaling molecules that have immunomodulatory biological activities. They may either be pro-inflammatory or anti-inflammatory. Pro-inflammatory cytokines include TNF-α, IFN-γ, and IL-1β [7]. Normal levels of inflammatory cytokines are essential for cell signaling and a healthy protective immune response to pathogens, but long-term chronic inflammation caused by excessive stress and excessive production of inflammatory factors can cause tissue and cell damage [40,41]. Studies have shown that IL-1β can stimulate the production of some neurotoxic proteins, inhibit the secretion of neuroprotective factors, have direct toxic effects on nerve cells, and inhibit the regeneration of neurons. Additionally, excessive IL-1β overexpression can over-activate the hypothalamus-pituitary-adrenal (HPA) axis, leading to raised glucocorticoid levels [42,43,44]. Elevated levels of TNF-α can activate 5-HT transporters and promote pre-synaptic reuptake of 5-HT, and this can lead to depression caused by a decrease in 5-HT levels in the synapse [45]. A high concentration of IFN-γ can aggravate the oxidative stress injury of local neurons, have a toxic effect on neural stem cells, and change the growth and differentiation state of neural stem cells [46,47].

In addition, studies have shown that pro-inflammatory cytokines such as TNF-α, IL-1β, and IFN-γ are effective activators of NF-κB and IDO [48,49,50].

The NF-κB signaling pathway plays an important role in depression-like behavior induced by acute and chronic stress and LPS [51,52]. NF-κB is closely related to neuroinflammation: On one hand, NF-κB, as a multi-effect regulator, is involved in the regulation of inflammatory mediators and the transcription and expression of inflammatory cytokines [53]; on the other hand, inflammatory factors such as TNF-α, IL-1β, and IFN-γ can activate NF-κB. This cycle aggravates the inflammatory response. Our results show that honokiol can effectively reverse the increase in NF-κB mRNA and protein expression in the hippocampi of LPS-treated depression model mice, suggesting that the improvement of LPS-induced depression-like behavior by honokiol may be related to the inhibition of NF-κB activation.

The ‘5-HT depression hypothesis’ suggests that immune-mediated activation of pro-inflammatory cells induces IDO activation, leading to decreased plasma tryptophan levels and increased synthesis of harmful tryptophan catabolic metabolites, both of which synergize to aggravate the development of depression [54]. IFN-γ is a major inducer of IDO activation, while other pro-inflammatory cytokines such as TNF-α and IL-1β synergistically induce the activation of IDO [55,56,57]. IDO, a rate-limiting enzyme involved in tryptophan metabolism, is widely distributed in various tissues and participates in TRP–KYN metabolism. Therefore, the activity of IDO plays a crucial role in depression. The KYN/TRP ratio is often used as an activity index to evaluate IDO activity: The higher the ratio, the greater the activity of IDO. Disordered activation of IDO causes excessive activation of the TRP–KYN pathway, depleting tryptophan, and degrading more TRP to KYN, resulting in insufficient synthesis of 5-HT [58,59]. Studies have shown that the development of depressive symptoms was significantly correlated with the ratio of the kynurenine/kynurenic acid because kynurenine is neurotoxic, whereas kynurenic acid is neuroprotective. Thus, the KYN/KYNA ratio is often used as a measure of neurotoxicity and neuroprotection: The greater the ratio, the greater the neurotoxicity, which reflects an increase in the neurotoxic potential [60,61,62]. In this study, the *IDO* gene and protein expression levels and the KYN/TRP and KYN/KYNA ratios were significantly increased in the model group compared with the control group. However, in the drug-administered group, the *IDO* gene and protein expression levels and the KYN/TRP and KYN/KYNA ratios were significantly decreased compared with those in the model group, indicating that the antidepressant effect of honokiol may also be related to inhibition of IDO activation and an increase in neuroprotective metabolites (Figure 7).

Calcium can be called a ‘life and death signal’ and plays an important role in a variety of biological processes. Calcium homeostasis is essential for the function of the central nervous system [63]. As a second messenger, free calcium is an important signal transduction factor, participating in many activities such as enzyme system activation, hormone secretion, nucleotide metabolism, cell proliferation, and synaptic plasticity [64,65]. Under normal circumstances, calcium ions are maintained at a steady level, ensuring a dynamic balance. Disturbing this balance leads to apoptosis, neuronal degeneration, and necrosis [66]. It is well known that NMDA receptors transmit important information through the conduction of calcium ions. NMDAR (*N*-methyl-d-aspartate receptor) is a specific receptor for the excitatory amino acid glutamate, which is widely distributed in neurons of the hippocampus and hypothalamus, and is involved in synaptic plasticity regulation and advanced neural activity [67]. NMDAR dysfunction is associated with schizophrenia, neurodegenerative diseases, etc. [68]. Disordered activation of IDO also leads to an increase in the synthesis of harmful TRP catabolites such as QUIN, resulting in an imbalance of the QUIN/KYNA ratio. Quinolinic acid is an agonist of the NMDAR. Excessive accumulation of quinolinic acid over-activates NMDA receptors and excess NMDA receptor activity leads to cytotoxic calcium overload and neuronal damage further aggravates the occurrence of depression [62,69]. In this study, the Ca^2+^ concentration in the honokiol group was significantly reduced compared with that in the model group, indicating that the anti-depressant effect of honokiol may, in part, be related to its ability to lower free calcium levels in brain tissue, thus inhibiting calcium overload.

Studies have shown that honokiol can cross the blood-brain barrier and induce neuroblastoma cell apoptosis [32]. However, honokiol does not induce insults to non-neoplastic cells. In vitro and in vivo studies have also shown that honokiol has no obvious toxicity to normal brain cells [70]. Considering previous studies on the toxicity of magnolia bark extract, its safety seems to be guaranteed [21,71]. However, it has recently been reported that honokiol interacts with aristolochic acid in vitro to enhance the toxicity of aristolochic acid, suggesting that honokiol may also interact with other compounds derived from plants [72]. Based on this, we should pay more attention to the side effects of honokiol in future research.

## 4. Materials and Methods

### 4.1. Animals and Ethical Review

Specific pathogen free (SPF) grade, adult male ICR mice, weighing 22–25 g, were provided by Beijing HuaFukang Biotechnology Co., Ltd., animal license number: SCXK (Beijing) 2014-0004. All animals were housed under the following standard conditions: A 12 h light/dark cycle; a humidity range of 40 ± 10%; and an ambient temperature of 22 ± 1 °C. The mice had free access to standard food and water. All experimental procedures involving the animals were approved by the Experimental Animal Ethics Committee of the Academic Committee of Beijing University of Chinese Medicine (project identification code: BUCM-4-2019011501-1006).

### 4.2. Animal Study Design

Thirty mice were randomly divided into a control group, an LPS group, and an LPS + Honokiol group (*n* = 10/group). Honokiol was suspended in 0.5% sodium carboxymethylcellulose (CMC-Na) at certain concentrations before use.

Studies have shown that pretreatment with honokiol (10 mg/kg) significantly improved depression-like behavior induced by chronic restraint stress [73]. In addition, in our previous study [35], we studied the antidepressant-like effects of honokiol on acute and chronic stress mice by setting up three dose groups (2.5, 5, and 10 mg/kg). Our results also showed that pretreatment with honokiol (10 mg/kg) significantly improved depression-like behavior.

Based on our previous experiments, honokiol was orally administered once daily for 11 days at a dose of 10 mg/kg body weight (bw), and the control and LPS groups were given an equal volume of 0.5% CMC-Na solution. Autonomic activity tests were performed 30 min after administration on Day 10, and on Day 11, the animals in the model and honokiol groups were injected intraperitoneally with 1 mg/kg bw LPS. Behavioral tests were performed 4 h later. After the behavioral tests, the eyeballs were removed for blood collection, the animals were killed by decapitation, the brain of each animal was removed and stored on ice for analysis.

### 4.3. Drugs and Reagents

Honokiol (batch number: P1095220) was obtained from Shanghai Titan Scientific Co., Ltd. (Shanghai, China). The structure of honokiol was shown in Figure 8. LPS (batch number: RH51487) was purchased from BioRuler (Connecticut, USA). IFN-γ (batch number: 228080442) ELISA kits were purchased from Multisciences (Lianke) Biotech, Co., Ltd. (Hangzhou, China). IL-1β (batch number: 20171012), TNF-α (batch number: 20171020), IFN-γ (batch number: 20171018) radioimmunoassay kits were purchased from Beijing Sino-UK Institute of Biological Technology. (Beijing, China). The Hipure Total RNA mini kit (No. R4111-02) was obtained from Magen Bio (Guangzhou, China). The Reveraid First Strand cDNA Synthesis kit (No. K1622) was from Thermo Scientific (San Diego, CA, USA); and the SYBR PCR master mix was from Invitrogen (San Diego, CA, USA).

The IDO1 antibody (No: 66528-1-Ig), NF-κB P65 antibody (No: 14220-1-AP), and the ECL chemiluminescence detection kit (No: B500022) were obtained from the Proteintech Group, Inc. (Chicago, IL. USA). The GAPDH antibody (No: cw0100) was obtained from Beijing Kangwei Century Biotechnology Co., Ltd. (Beijing, China). The calcium ion fluorescent probe Fluo-3-Am (batch number: KGAF023), PBS (No: KGB5001), RIPA buffer (No: KGP703-100), and the BCA protein quantitation kit were purchased from KeyGen BioTech Corp., Ltd. (Nanjing, China). The Minute™ Total Protein Extraction kit (animal cells/tissues) (No: SD-001/SN-002) was purchased from Invent Biotechnologies, Inc. (Eden Prairie, MN, USA). Foetal calf serum (No: hz001) was purchased from the Sijiqing Company Ltd. (Hangzhou, China), and collagenase type II (No: HA0890-100 mg) was purchased from Sigma-Aldrich (Saint Louis, MO, USA).

### 4.4. Behaviour Testing

#### 4.4.1. Autonomic Activity Tests

Autonomic activity tests are used to measure the general behavior of animals and to assess behavioral changes in rodents exposed to new environments. The tests were performed on the 10th day of dosing, 30 min after drug administration. The mice were placed in a self-made opaque 60 cm by 60 cm by 40 cm activity box with a defined central square area of 30 cm by 30 cm. A small animal behavior analysis system (Etho-Vision XT9, Noldus, The Netherlands), was used to record the distance, velocity, and frequency of movement of the mice across both the central and marginal regions over 5 min.

#### 4.4.2. Forced Swimming Test

This test is used to evaluate desperation in animals and is one of the classic models used to assess the efficacy of antidepressants [74]. The experimental animals were subjected to an FST 4 h after the LPS injection (see Appendix A). The mice were placed individually in transparent cylindrical glass cylinders (25 cm high and 10 cm in diameter) with a water depth and temperature of 10 cm and 23–25 °C, respectively. Opaque baffles were placed between the cylinders to prevent the mice from seeing each other. The activity of the mice was observed a period of over 6 min using a small animal behavior analysis system (Etho-Vision XT9), and the duration of immobility over the last 4 min of the 6 min was recorded. Mice were judged to be immobile when they stopped struggling and floated on the water or made only small movements necessary to keep its head above water.

#### 4.4.3. Tail Suspension Test

The tail suspension test is also one of the classic models used to assess the efficacy of antidepressants [75]. The experimental animals were subjected to a TST 4 h after the LPS injection. The end of the tail of each mouse (2 cm from the tip) was fixed with medical tape to a tension transducer (JZ100, Shanghai Kang Wei Medical Technology Development Co., Ltd., China) so that the mouse was hanging upside down with its head approximately 15 cm off the ground. The activity of the animals was observed over 6 min using a MedLab BioInformatics acquisition system (MedLab-U/4C501H, Nanjing Medease Science and technology Co., Ltd., China), and the immobility time over the last 4 min was recorded.

### 4.5. Analysis of Pro-Inflammatory Cytokine Levels

#### 4.5.1. Detection of TNF-α, IL-1β, and IFN-γ in Serum

After behavioral tests, the eyeballs were taken for blood collection and 300 µL samples of the supernatant were removed for radioimmunoassay. The concentration of each pro-inflammatory cytokine in the serum samples was determined by radioimmunoassay kits according to the manufacturer’s instructions.

#### 4.5.2. Detection of IFN-γ in Brain Tissue

After the steps detailed in Section 4.5.1, mice were killed and their brains were rapidly removed, and the hippocampus and cerebral cortex were isolated. The cerebral cortex was placed in a centrifuge tube and 4 °C saline was added at four times the volume of tissue. The tissue was thoroughly homogenized by an electric homogenizer and placed on ice. The IFN-γ content in the supernatant was determined according to the ELISA kit instructions.

### 4.6. RT–PCR Detection of Hippocampal IDO and NF-κB P65 Gene Expression

The hippocampus was dissected from the brain tissue, and the total RNA was extracted according to the Hipure Total RNA mini kit instructions. The RNA concentration was measured using an ultraviolet spectrophotometer (UV-2000, Unico, Shanghai, China). Reverse transcription was performed on a T100 Thermal Cycler PCR machine (Bio-Rad, USA) using the Reveraid First Strand cDNA Synthesis kit and the SYBR PCR master mix according to the manufacturers’ instructions. Amplification and quantitative detection were performed in a Real-Time PCR machine (Bio-Rad, USA). The RT–PCR protocol was as follows: initial denaturation at 95 °C for 15 min, then 55 cycles at 95 °C for 10 s, and finally 58 °C for 30 s. Primer sequences for the genes of interest were as follows:
IDO1 F: 5′ GGATCCTTGAAGACCACCACAT 3′IDO1 R: 5′ AAGGACCCAGGGGCTGTAT 3′NF-κB P65 F: 5′ ATCATCGAACAGCCGAAGCA 3′NF-κB P65 R: 5′ TGATGGTGGGGTGTGTCTTG 3′β-actin F: 5′ CCTAGGCACCAGGGTGTG 3′β-actin R: 5′ CGGTGAGCAGCACAGGGT 3′The 2^−ΔΔ*C*t^ method was used for relative quantitative analysis of the results.


### 4.7. Western Blot Detection of Hippocampal IDO and NF-κB P65 Protein Expression

The hippocampus sample was obtained as in Section 4.5.2. RIPA buffer was added for tissue lysis and extraction of total protein. Protein concentration determination was performed using a BCA kit, and the samples were separated by electrophoresis on a 5–10% polyacrylamide gel. After separation, the proteins were transferred to a 0.45 μm PVDF membrane by the wet transfer method using constant piezoelectricity at 100 V for 1.5 h. The membrane was blocked in 5% skimmed milk for 2 h, incubated overnight with the primary antibody, washed in a TBST solution (SolarBio, Beijing, China) for 3 by 10 min, incubated in the secondary antibody at room temperature for 1 h on the shaker, washed a further three times in TBST solution, and placed on plastic wrap. Developer from the ECL chemiluminescence detection kit was added evenly, and the membrane was allowed to stand. The filter was blotted dry for 1 min, and an Azure multifunctional molecular imaging system (C600, Azure, USA) was used for detection of the labelled proteins. GAPDH was used as the control for protein loading and transfer efficiency Gray value analysis was performed using Image J software (The National Institutes of Health, Bethesda, MD, USA).

### 4.8. HPLC–MS Detection of TRP–KYN Metabolites

The serum content of 5-HT, KYN, TRP, KYNA, and QUIN was determined by HPLC–MS. The IDO activity is represented by the KYN/TRP ratio, and the neuroprotective effect by the KYN/KYNA ratio. HPLC analysis was performed using a DIONEX Ultimate 3000 HPLC system (Thermo Fisher Scientific, USA). The liquid phase conditions used were: Column: MSLab C18 (150 by 4.6 mm 5 µm); mobile phase, A aqueous phase: Water (1% formic acid); B organic phase: acetonitrile (1% formic acid); injection volume: 5 µL; column temperature: 50 °C; flow rate: 1 mL/min; and gradient elution: 0–1 min, (10% B); 1–12 min, (10%–70% B); 12–12.1 min, (70%–100% B); 12.1–15 min, (100% B); 15–15.1 min, (100%–10% B); and 15.1–20 min, (10% B). Mass spectrometry was performed using an API 3200 QTRAP system (AB SCIEX, USA). The mass spectrometry conditions used were: Scanning mode: Multiple reaction monitoring (MRM); ion source: Electron spray ionization (ESI); atomization temperature (TEM): 500 °C; injection voltage (EP): 10; spray voltage (IS): +5500 V; collision chamber injection voltage (CXP): 2.0; atomizing gas (GS1): 55 psi; collision gas (CAD): medium; assist gas (GS2): 60 psi; and air curtain gas (CUR): 20 psi.

### 4.9. Determination of Free Ca^2+^ Concentration

After the brain was removed from the animals, the surface of the tissue was washed with PBS and the tissue was placed in a clean culture dish. The membrane on the tissue surface was removed with a cotton ball, and the tissue was washed in PBS and diced. A volume of 1 mL of 0.4% collagenase solution prepared in PBS was added to the dish, and the contents were transferred to a 15 mL centrifuge tube and made up to 2 mL with 0.4% collagenase. The sample was incubated at 37 °C for 40 min, shaking once every 10 min, and the reaction was terminated by adding 3 mL of 10% fetal calf serum. The solution was mixed, filtered through a 200 mesh, and centrifuged at 670× *g* for 5 min at 4 °C. The supernatant was discarded and the pellet was resuspended in 5 mL PBS. A volume of 1 mL cell suspension (approximately 10^7^ cells) was centrifuged again at 670× *g* for 5 min at 4 °C, and the supernatant was discarded. The cells were resuspended in 1 mL of PBS, Fluo-3-Am was added to a final concentration of 5 μM, and the solution was incubated at 37 °C for 30 min then rinsed twice with PBS to remove excess dye. The cells were resuspended and 300 μL of the solution was filtered into a flow tube, and the free calcium ion concentration was measured using a flow cytometer (FACS Canto II, BD, USA).

### 4.10. Statistical Analysis

Data were expressed as mean ± SEM, and were analyzed by one-way ANOVA followed by Dunnett’s multiple comparisons test. Statistical analysis was performed using SAS 8.2 (IBM, Armonk, NY, USA) and GraphPad Prism 6.01 (GraphPad Software Inc, San Diego, CA, USA). *p* < 0.05 was considered statistically significant.

## 5. Conclusions

Our results indicate that honokiol exhibits significant antidepressant-like effects and can improve depression-like behavior in LPS-induced depression mice. Its main mechanism of action may be to inhibit the activation of NF-κB, reduce the level of IFN-γ in the brain, and pro-inflammatory cytokines such as TNF-α, IFN-γ, and IL-1β in the peripheral blood, thereby inhibiting the inflammatory response. Additionally, it also affects tryptophan metabolism, reducing the expression of a key enzyme of tryptophan metabolism, IDO. This anti-depression-like effect is also associated with inhibition of intracellular calcium overload and increased neuroprotective metabolites.

## Figures and Tables

**Figure 1 molecules-24-02035-f001:**
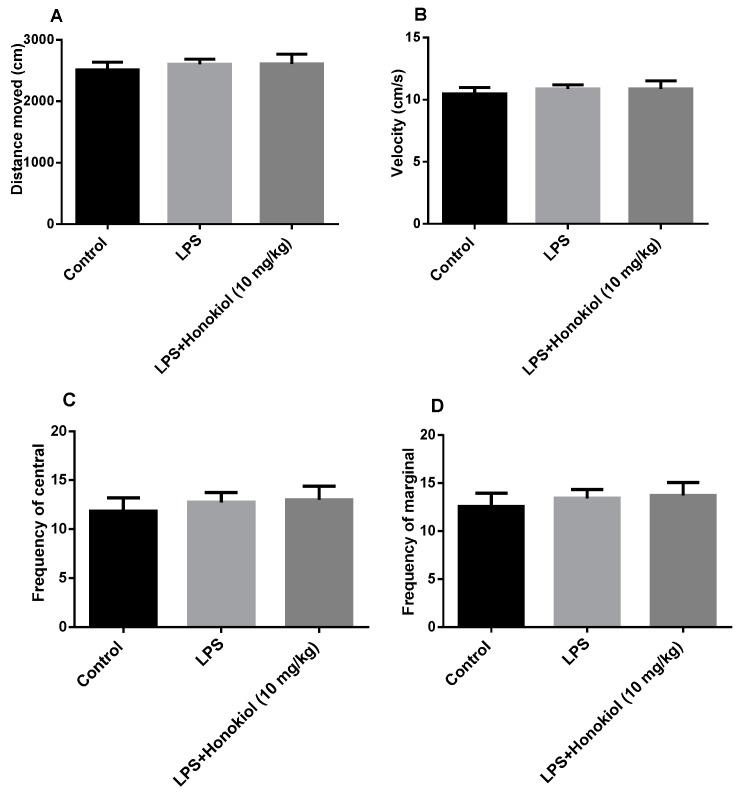
The effects of lipopolysaccharide (LPS) and honokiol on autonomous activity. (**A**) The distance moved. (**B**) The velocity of movement. (**C**) The central region dwelling frequency. (**D**) The marginal region dwelling frequency. Data are expressed as mean ± SEM (*n* = 10/group). The results showed no difference between the different groups.

**Figure 2 molecules-24-02035-f002:**
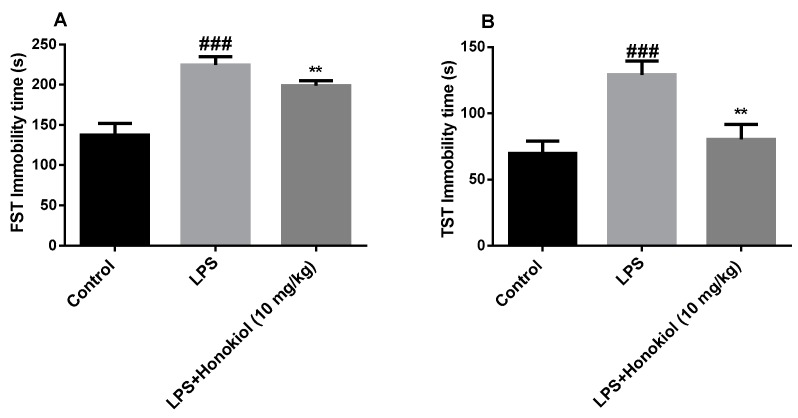
The effects of honokiol on (**A**) the forced swim test (FST) and (**B**) the tail suspension test (TST). Data are expressed as mean ± SEM (*n* = 10/group). ^###^
*p* < 0.001 LPS vs. control; ^**^
*p* < 0.01 LPS + Honokiol vs. LPS.

**Figure 3 molecules-24-02035-f003:**
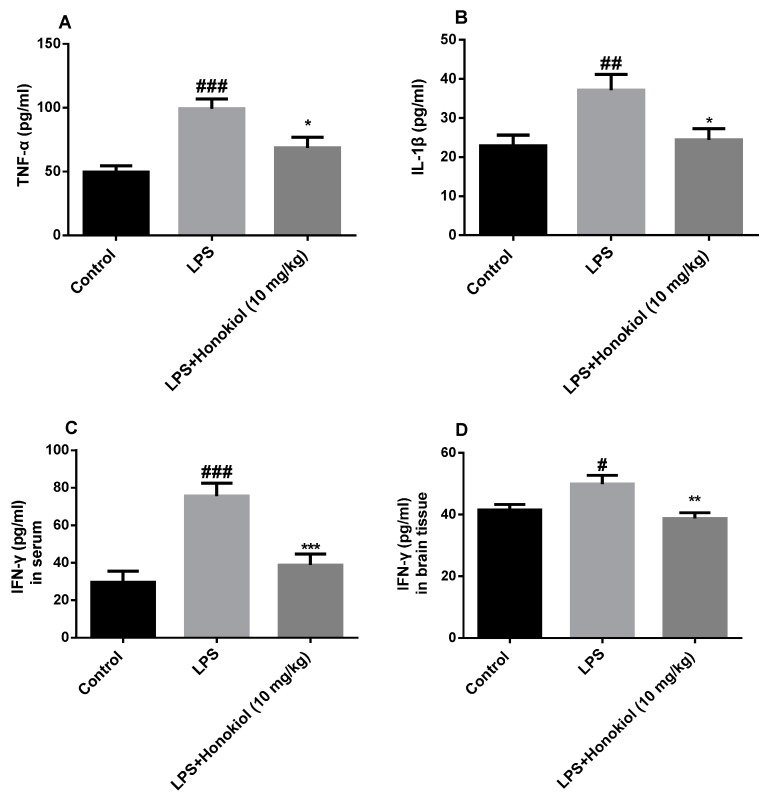
The effects of honokiol on proinflammatory cytokines in serum and in brain tissue. Serum: (**A**) tumor necrosis factor α (TNF-α), (**B**) interleukin 1β (IL-1β), and (**C**) interferon γ (IFN-γ); brain tissue: (**D**) IFN-γ. Data are expressed as mean ± SEM (*n* = 9). ^###^
*p* < 0.001, ^##^
*p* < 0.01, ^#^
*p* < 0.05 LPS vs. control; *** *p* < 0.001, ** *p* < 0.01, * *p* < 0.05 LPS + Honokiol vs. LPS.

**Figure 4 molecules-24-02035-f004:**
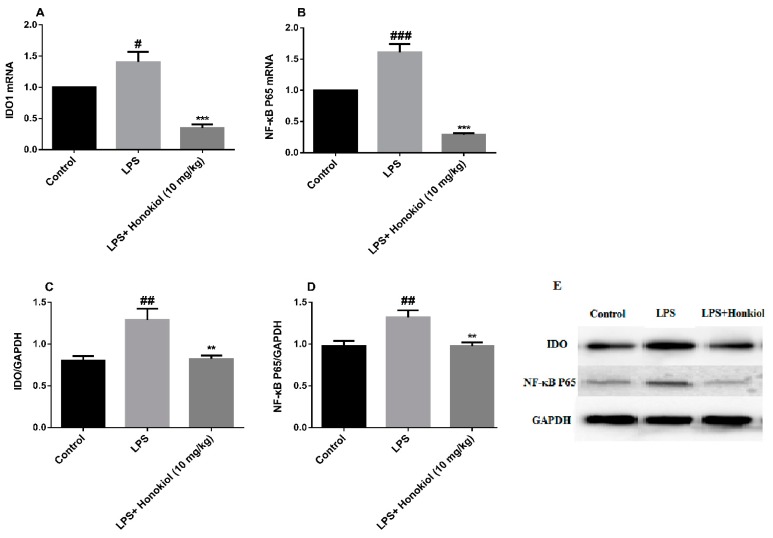
The effects of honokiol on mRNA and protein expression of hippocampal indoleamine 2,3-dioxygenase (IDO) and NF-κB. (**A**) IDO1 mRNA expression; (**B**) NF-κB mRNA expression; (**C**) IDO protein expression; (**D**) NF-κB protein expression; (**E**) The western blot bands. Data are expressed as mean ± SEM (*n* = 8). ^###^
*p* < 0.001, ^##^
*p* < 0.01, ^#^
*p* < 0.05 LPS vs. control; *** *p* < 0.001, ** *p* < 0.01 LPS + Honokiol vs. LPS.

**Figure 5 molecules-24-02035-f005:**
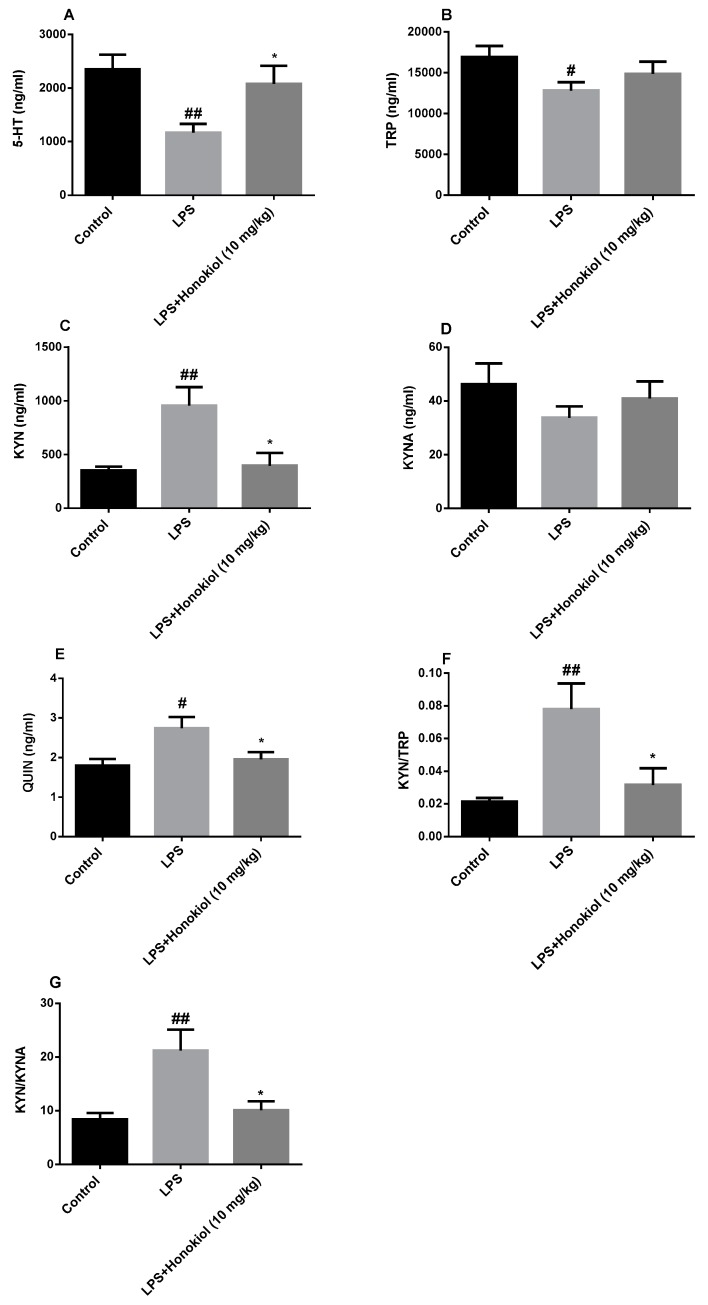
The effects of honokiol on the metabolites of the tryptophan–kynurenine (TRP–KYN) pathway. (**A**) serotonin (5-HT); (**B**) TRP; (**C**) KYN; (**D**) kynurenic acid (KYNA); (**E**) quinolinic acid (QUIN); (**F**) KYN/TRP ratio; and (**G**) KYN/KYNA ratio. Data are expressed as mean ± SEM (*n* = 5). ^##^
*p* < 0.01, ^#^
*p* < 0.05 LPS vs. control; * *p* < 0.05 LPS + Honokiol vs. LPS.

**Figure 6 molecules-24-02035-f006:**
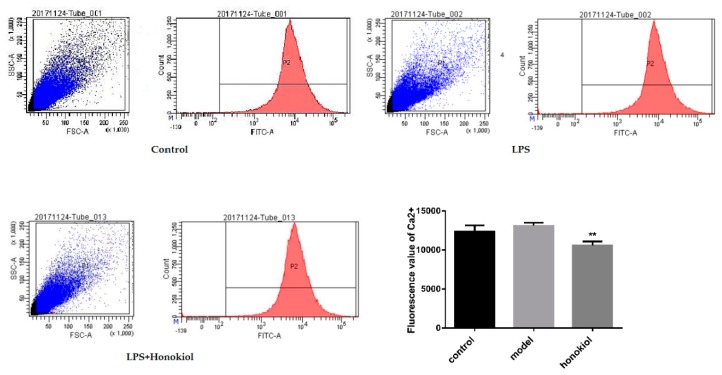
The effect of honokiol on free Ca^2+^ levels in the brain cells of mice. Data are expressed as mean ± SEM (*n* = 4). ** *p* < 0.01 LPS + Honokiol vs. LPS.

**Figure 7 molecules-24-02035-f007:**
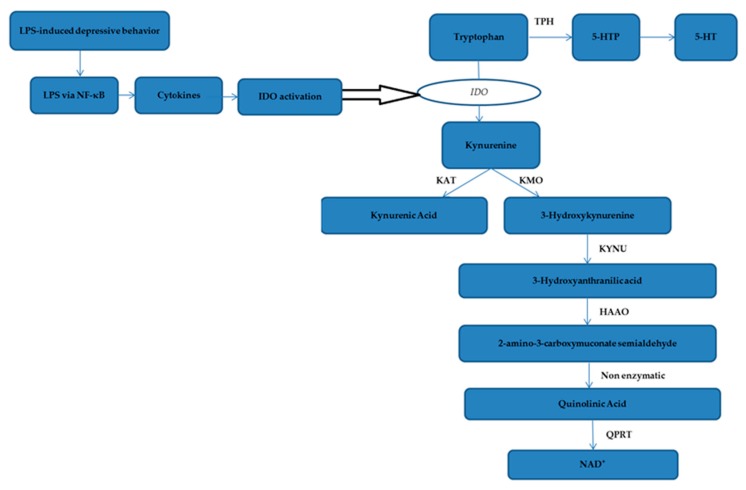
IDO pathway and tryptophan metabolism pathway. Notes: Tryptophan catabolism through the kynurenine pathway involves IDO as an intracellular rate–limiting enzyme. Pro-inflammatory cytokine is upregulated by LPS regulated by NF-κB. Activation of NF-κB results in the release of pro-inflammatory cytokines, which in turn, activates IDO activity. Tryptophan may be metabolized to serotonin or alternatively, is metabolized via the kynurenine pathway. Tryptophan is metabolized to kynurenine by indoleamine 2,3-dioxygenase. Kynurenine is then converted via kynurenine aminotransferases (KAT) to kynurenic acid, a neuroprotective molecule as it antagonizes glutamate receptor-induced neurotoxicity. In addition, kynurenine can also be converted to 3-hydroxykynurenine by kynurenine-3-monooxygenase (KMO), for which evidence is accumulating of its neurotoxic capability. Sequential conversion to 2-amino-3-carboxymuconate-semialdehyde is the penultimate step leading to enzymatic production of (neuroprotective) picolinic acid, and the (non-enzymatic) production of the well-known neurotoxic compound quinolinic acid. Further conversion of QUIN to the essential cofactor NAD+ is catalysed by quinolinate phosphoribosyltransferase (QPRT).

**Figure 8 molecules-24-02035-f008:**
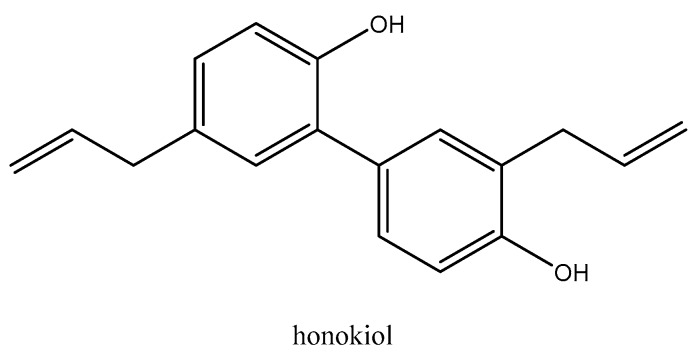
The structure of honokiol (Chemical formula: C_18_H_18_O_2_; Molecular weight: 266.34).

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
