# Peer review of "Antidepressant-Like Effect and Mechanism of Action of Honokiol on the Mouse Lipopolysaccharide (LPS) Depression Model"

_molecules, 2019, doi:10.3390/molecules24112035_

Reviewer 1 Report

Authors extend characterization of biological properties of the honakiol. The scientific hypothesis is interesting, and defined as anti-inflammatory properties of honakiol improve depression-like behavior of animals in FSt and TST. To support the hypothesis authors determined several inflammation markers using in vitro methods. Especially connection the obtained results with KYN/KYNA system is interesting. So, the work merits publication in Molecules, upon providing additional comments and/or results:

1) why honakiol was tested in vivo only in one dose? To determine the pharmacological properties it is highly recommended to test at least 3 doses - such approach would enable verification if there is a dose-dependance or only one dose is active

2) a positive positive, active in LPS setting, shall be used in FST and TST

3) Figure 4: may Authors explain if the observed decrease of expression of IDO and NK-kB P65 below a control is positive or it might stimulate any re-compensation response  

4)  please explain in text abbreviation IOD1

5) please introduce a figure showing structure of honokiol.

I recommend to accept after major revision.

Author Response

Dear Editors and Reviewers:

Thank you for your letter and for the reviewers’ comments concerning our manuscript entitled “Antidepressant-like effect and mechanism of action of honokiol on the mouse lipopolysaccharide (LPS) depression model” (ID: molecules-498372). Those comments are all valuable and very helpful for revising and improving our paper, as well as the important guiding significance to our researches. We have studied comments carefully and have made correction which we hope meet with approval. Revised portion are marked in red in the paper. The main corrections in the paper and the responds to the reviewer’s comments are as flowing:

Response to Reviewer 1 Comments:

1)       why honakiol was tested in vivo only in one dose? To determine the pharmacological properties it is highly recommended to test at least 3 doses - such approach would enable verification if there is a dose-dependance or only one dose is active

Response:

First of all, thank you very much for your fruitful comments. As you said, setting different doses is crucial.

In our previous study, we set up three dose groups (2.5, 5, 10 mg/kg ) to study the antidepressant effect and mechanism of honokiol on acute and chronic stress mice. Our results indicate that honokiol can improve the depression behavior of mice and has certain anti-depression effects. The main mechanism of action is related to the decrease of IDO content and gene expression level of tryptophan pathway and increase of 5-HT level in the brain. And we found that the high dose group of magnolol (10 mg/kg) had the most significant effect. Based on this, we carried out this experiment. I hope my answer can eliminate your doubts. Attached to our published article.

34.     Wang, P. P., Liu, B. X., Yang, T., Li, L. N., Wang, S. N., Chang, H.S. Antidepressant effect and mechanism of honokiol on acute and chronic stress mouse. Chinese Pharmaceutical Journal 2017, 52, (24), 2161-2165.

2) a positive positive, active in LPS setting, shall be used in FST and TST

Response:

Thank you very much for your fruitful comments.

Here I would like to explain the situation. In the previous experimental study, in order to verify the antidepressant-like effect of honokiol, fluoxetine was set as a positive drug for comparison (Figure 1). In addition, the fluoxetine group was designed at the beginning of this experiment, and the results are as follows (Figure 2). Both experimental results showed that honokiol had an antidepressant-like effect similar to fluoxetine.

This experiment is mainly to explore the possible mechanism of the antidepressant-like effect of honokiol, so fluoxetine was not tested separately in the later experiments.

Here I enclose our published articles for your reference.

34.     Wang, P. P., Liu, B. X., Yang, T., Li, L. N., Wang, S. N., Chang, H.S. Antidepressant effect and mechanism of honokiol on acute and chronic stress mouse. Chinese Pharmaceutical Journal 2017, 52, (24), 2161-2165.

3) Figure 4: may Authors explain if the observed decrease of expression of IDO and NK-kB P65 below a control is positive or it might stimulate any re-compensation response

Response:

Thank you very much for your fruitful comments.

Here I must admit that the expression of IDO and NK-kB P65 gene is lower than that of normal group. In this experiment, the results of HPLC-MS and protein expression did not decrease compared with the normal group, but inhibited gene expression. It is suggested that the effect of honokiol on the expression of related genes may involve more complex mechanisms.

So, further studies are clearly necessary to fully evaluate the reasons why honokiol causes the expression of IDO and NK-kB P65 to decrease below the control. This will also be an important part of our next experiment. I hope that you can understand the technical problems we face and thank you again for your suggestions.

4)  please explain in text abbreviation IDO 1

Response:

the statements of “IDO 1” were corrected as “IDO 1(indolamine-2,3-dioxygenase 1)” ( New manscriptLine 130).

IDO mainly includes IDO1 and IDO2, and IDO1 has a higher affinity for TRP than IDO2. In humans, IDO1 has an evolutionary paralog (indolamine-2,3-dioxygenase 2; IDO2) and a functional ortholog (tryptophan-2,3-dioxygenase; TDO) that catalyze the same biochemical reaction; however, IDO2 and TDO show high tissue specificity and much lower expression level than IDO1 that significantly restrict their activity[1, 2].

Tryptophan (trp) catabolism by indolamine-2,3-dioxygenase 1 (IDO1) is a chief endogenous metabolic pathway that tightly regulates unwanted immune responses through consumption of trp and generation of immunosuppressive kynurenines (kyn)[3].

1.Bilir, C.; Sarisozen, C., Indoleamine 2,3-dioxygenase (IDO): Only an enzyme or a checkpoint controller? Journal of Oncological Sciences 2017, 3, (2), 52-56.

2.van Baren, N.; Van den Eynde, B. J., Tryptophan-Degrading Enzymes in Tumoral Immune Resistance. Frontiers in Immunology 2016, 6, (34).

3.Lanz, T. V.; Opitz, C. A.; Ho, P. P.; Agrawal, A.; Lutz, C.; Weller, M.; Mellor, A. L.; Steinman, L.; Wick, W.; Platten, M., Mouse Mesenchymal Stem Cells Suppress Antigen-Specific TH Cell Immunity Independent of Indoleamine 2,3-Dioxygenase 1 (IDO1). Stem Cells and Development 2009, 19, (5), 657-668.

5) please introduce a figure showing structure of honokiol.

Response:

Thank you very much for your fruitful comments. According to your suggestion, I added the structure diagram of honokiol. As shown in the following figure: ( New manscriptLine 303-306).

Figure 8. The structure of honokiol (Chemical formula: C18H18O2; Molecular weight: 266.34).

Thank you again for your fruitful comments.

We tried our best to improve the manuscript and made some changes in the manuscript. These changes will not influence the content and framework of the paper. And here we did not list the changes but marked in yellow in revised paper.

We appreciate for your warm work earnestly, and hope that the correction will meet with approval.

Once again, thank you very much for your comments and suggestions.

Sincerely,

Bo Zhang

Reviewer 2 Report

page 1 abstract: it is too long

page 3 line 96: correct “testes”

page 5 line 140: change to “is involved”

page 5 line 141: correct “this models”

page 8 figures 9-11 caption: detect? They should be revised

page 9 lines 246 and 248: I suggest italicizing the gene name

page 12 line 346: change to “were placed

page 12 line 377: missing dot

page 13 lines 402 and 404: please convert “rpm” in “g”

page 13 lines 420-421: please revise the sentence

Author Response

Dear Editors and Reviewers:

Thank you for your letter and for the reviewers’ comments concerning our manuscript entitled “Antidepressant-like effect and mechanism of action of honokiol on the mouse lipopolysaccharide (LPS) depression model” (ID: molecules-498372). Those comments are all valuable and very helpful for revising and improving our paper, as well as the important guiding significance to our researches. We have studied comments carefully and have made correction which we hope meet with approval. Revised portion are marked in red in the paper. The main corrections in the paper and the responds to the reviewer’s comments are as flowing:

Response to Reviewer 2 Comments:

1) page 1 abstract: it is too long

Response:

Thank you very much for your fruitful suggestions.We have rewritten this part according to your suggestion, please check it out.

Original manuscript page 1 line 10-38 were corrected as follows (New manuscript page 1 line 10-28):

There is growing evidence that neuroinflammation is closely linked to depression. Honokiol, a biologically active substance extracted from Magnolia officinalis, which is widely used in the traditional Chinese medicine, has been shown to exert significant anti-inflammatory effects and improve depression-like behaviour caused by inflammation. However, the specific mechanism of action of this activity is still not clear. In this study, the lipopolysaccharide (LPS) mouse model was used to study the effect of honokiol on depression-like behaviour induced by LPS in mice and its potential mechanism. A single administration of LPS (1 mg/kg, i.p.) increased the immobility time in the forced swimming test (FST) and tail suspension test (TST), without affecting autonomous activity. Pretreatment with Honokiol (10 mg/kg, p.o.) for 11 consecutive days significantly improved the immobility time of depressed mice in the FST and TST experiments. Moreover, honokiol ameliorated LPS-induced NF-κB activation in the hippocampus and significantly reduced the levels of the proinflammatory cytokines(TNF-α, IL-1β and IFN-γ ). In addition, honokiol inhibited LPS-induced indoleamine 2, 3-dioxygenase (IDO)  activation and quinolinic acid (a toxic product) increase, and reduced the level of free calcium in brain tissue, thereby inhibiting calcium overload. In summary, our results indicate that the anti-depressant-like effects of honokiol are mediated by its anti-inflammatory effects. Honokiol may inhibit the LPS-induced neuroinflammatory response through the NF-κB signalling pathway, reducing the levels of related pro-inflammatory cytokines, and furthermore, this may affect tryptophan metabolism and increase neuroprotective metabolites.

2) page 3 line 96: correct “testes”

Response:

We are very sorry for our incorrect writing.

Original manuscript page 3 line 96, the “testes” were corrected as “tests” (New manuscript page 2 line 86)

3) page 5 line 140: change to “is involved”

Response:

Original manuscript Line page 5 line 140, the “are involved” were corrected as “is involved” (New manuscript page 5 line 129).

4) page 5 line 141: correct “this models”

Response:

Original manuscript page 5 line 141, “in this models” was deleted

5) page 8 figures 9-11 caption: detect? They should be revised

Response:

Original manuscript page 8 figures 9-11 caption, the “Flow cytometry detect” was deleted.

Figures 9-11 is now integrated into a single diagram (New manuscript page 7 line 168).

6) page 9 lines 246 and 248: I suggest italicizing the gene name

Response:

Original manuscript page 9 lines 246 and 248, the “IDO” were corrected as “IDO” (New manuscript page 8 line 218-220).

7) page 12 line 346: change to “were placed”

Response:

Original manuscript page 12 line 346, the “was placed” were corrected as “were placed” (New manuscript page 11 line 344).

8) page 12 line 377: missing dot

Response:

Original manuscript page 12 line 377 ,“.” was added (New manuscript page 12 line 375).

9) page 13 lines 402 and 404: please convert “rpm” in “g”

Response:

Original manuscript page 13 lines 402 and 404, the “rpm” were corrected as “g” (New manuscript page 12 line 400-402).

10) page 13 lines 420-421: please revise the sentence

Response:

Original manuscript page 13 lines 420-421, the statements of “Additionally, it also affecting tryptophan metabolism, reducing the expression of a key enzyme of tryptophan metabolism, IDO, and inhibits intracellular calcium overload increasing neuroprotective metabolites.” were corrected as “Additionally, it also affecting tryptophan metabolism, reducing the expression of a key enzyme of tryptophan metabolism, IDO. And this anti-depression-like effect is also associated with inhibition of intracellular calcium overload and increased neuroprotective metabolites” (New manuscript page 13 line 418-421).

Thank you again for your fruitful comments.

We tried our best to improve the manuscript and made some changes in the manuscript. These changes will not influence the content and framework of the paper. And here we did not list the changes but marked in yellow in revised paper.

We appreciate for your warm work earnestly, and hope that the correction will meet with approval.

Once again, thank you very much for your comments and suggestions.

Sincerely,

Bo Zhang

Reviewer 3 Report

This manuscript aimed to investigate the anti-depressant-like effects of honokiol against LPS-induced neuroinflammation and the capacity of honokiol to combat depression. Despite several studies have pointed out that honokiol effectively abrogated depressive disorder, the authors here provide insight into potential mechanisms including tryptophan metabolism and its neuroprotective metabolites. Accordingly, the current study extends the latent application of honokiol as a complementary and alternative antidepressant. However, there are some points should be addressed before it is suitable for publication.

Minor comments:

1. According to the structure of honokiol (5,3’-Diallyl-2,4-dihydroxybiphenyl) and its ability to cross the blood-brain barrier (BBB), honokiol should have a  hydrophorbic property. It would be appropriate to address the vehicle used in honokiol group. 

2. Accumulating evidence reported that Magnolia mixture and honokiol possess hepatotoxic and induce apoptosis in neuroblastoma cells (Poivre and Duez, J Zhejiang Univ-Sci B Biomed & Biotechnol, 2017; Lin et al., Neuro-Oncology 2012). Thus, the side-effects of honokiol should be expected. There was no honokiol alone group and neither histological analysis to demonstrate the safety of honokiol. The authors should provide discussion regarding the safety issue.

3. Page 3, line 96: “testes” is misspelled.

4. Page 13, line 407: “The cells were resuspended 300 μl of the solution was transferred into a flow tube,” This sentence is incorrect.

5. References: The journal titles are missing from several cited references.

Author Response

Dear Editors and Reviewers:

Thank you for your letter and for the reviewers’ comments concerning our manuscript entitled “Antidepressant-like effect and mechanism of action of honokiol on the mouse lipopolysaccharide (LPS) depression model” (ID: molecules-498372). Those comments are all valuable and very helpful for revising and improving our paper, as well as the important guiding significance to our researches. We have studied comments carefully and have made correction which we hope meet with approval. Revised portion are marked in red in the paper. The main corrections in the paper and the responds to the reviewer’s comments are as flowing:

Response to Reviewer 3 Comments:

Minor comments:

1. According to the structure of honokiol (5,3’-Diallyl-2,4-dihydroxybiphenyl) and its ability to cross the blood-brain barrier (BBB), honokiol should have a  hydrophorbic property. It would be appropriate to address the vehicle used in honokiol group.

Response:

We are very sorry for our negligence. As you said, the honokiol is hydrophobic, we apologize for the irregularity and impreciseness of the previous manuscript. Based on your comments, we have made the following amendments (New manuscript page 9 line 276-279).

Honokiol was suspended in 0.5% sodium carboxymethylcellulose (CMC-Na) at certain concentrations before use. Based on our previous experiments, honokiol was orally administered once daily for 11 days at a dose of 10 mg/kg bw, and the control and LPS groups were given an equal volume of 0.5% CMC-Na solution.

2. Accumulating evidence reported that Magnolia mixture and honokiol possess hepatotoxic and induce apoptosis in neuroblastoma cells (Poivre and Duez, J Zhejiang Univ-Sci B Biomed & Biotechnol, 2017; Lin et al., Neuro-Oncology 2012). Thus, the side-effects of honokiol should be expected. There was no honokiol alone group and neither histological analysis to demonstrate the safety of honokiol. The authors should provide discussion regarding the safety issue.

Response:

Thank you very much for your fruitful comments. Based on your suggestions, we have added a brief discussion as follows (New manuscript page 8 line 243-250).

Studies have shown that honokiol can cross the BBB and induce neuroblastoma cell apoptosis [1]. However, honokiol does not induce insults to non-neoplastic cells. In vitro and in vivo studies have also shown that honokiol has no obvious toxicity to normal brain cells [2]. Considering previous studies on the toxicity of Magnolia bark extract, its safety seems to be guaranteed [3, 4]. However, it has recently been reported that honokiol interacts with aristolochic acid in vitro to enhance the toxicity of aristolochic acid, suggesting that honokiol may also interact with other compounds derived from plants [5].

Based on this, we should pay more attention to the side effects of honokiol in future research.

1. Lin, J.-W.; Chen, J.-T.; Hong, C.-Y.; Lin, Y.-L.; Wang, K.-T.; Yao, C.-J.; Lai, G.-M.; Chen, R.-M., Honokiol traverses the blood-brain barrier and induces apoptosis of neuroblastoma cells via an intrinsic bax-mitochondrion-cytochrome c-caspase protease pathway. Neuro-Oncology 2012, 14, (3), 302-314.

2. Lin, C.-J.; Chang, Y.-A.; Lin, Y.-L.; Liu, S. H.; Chang, C.-K.; Chen, R.-M., Preclinical effects of honokiol on treating glioblastoma multiforme via G1 phase arrest and cell apoptosis. Phytomedicine 2016, 23, (5), 517-527.

3. Sarrica, A.; Kirika, N.; Romeo, M.; Salmona, M.; Diomede, L., Safety and Toxicology of Magnolol and Honokiol. Planta Med 2018, 84, (16), 1151-1164.

4. Poivre, M.; Duez, P., Biological activity and toxicity of the Chinese herb Magnolia officinalis Rehder & E. Wilson (Houpo) and its constituents. Journal of Zhejiang University-SCIENCE B 2017, 18, (3), 194-214.

5. Bunel, V.; Antoine, M.-H.; Stévigny, C.; Nortier, J.; Duez, P., New in vitro insights on a cell death pathway induced by magnolol and honokiol in aristolochic acid tubulotoxicity. Food and Chemical Toxicology 2016, 87, 77-87.

3. Page 3, line 96: “testes” is misspelled.

Response:

We are very sorry for our incorrect writing .

Original manuscript Line 60-61, the “testes” were corrected as “tests” (New manuscript page 2 line 86).

4. Page 13, line 407: “The cells were resuspended 300 μl of the solution was transferred into a flow tube,” This sentence is incorrect.

Response:

We are very sorry for our negligence.

Original manuscript Page 13, line 407, the statements of “The cells were resuspended 300 μl of the solution was transferred into a flow tube,” were corrected as “The cells were resuspended and 300 μl of the solution was filtered into a flow tube,” (New manuscript page 12 line 405-406).

5. References: The journal titles are missing from several cited references.

Response:

We are very sorry for our incorrect writing.

All the places without the names of the journals have been corrected and marked red in the references. You can view it in the newly uploaded manuscript.

Thank you again for your fruitful comments.

We tried our best to improve the manuscript and made some changes in the manuscript. These changes will not influence the content and framework of the paper. And here we did not list the changes but marked in yellow in revised paper.

We appreciate for your warm work earnestly, and hope that the correction will meet with approval.

Once again, thank you very much for your comments and suggestions.

Sincerely,

Bo Zhang

Reviewer 4 Report

This study is largely based on tryptophan metabolism and kynurenine/tryptophan pathway, however the authors overlooked to provide a detailed explanation of their target mechanisms.   For example, what is the rationale for showing IDO and NF-kB p64 together (Figures 4, 5, and 6)? How are they associated? Based on the discussion (Line 236), I assume that NF-kB p64 acts as a transcription factor to promote IDO? In that case, the authors should show downregulation of IDO with NF-kB p65 inhibition. Also, there are multiple subunits and regulators in the NF-kB pathway. Why did the authors choose to investigate NF-kB p65 in particular? The target mechanisms introduced in the Line 65-72 must be specified, rather than using ambiguous terms such as ‘close relationship’ and ‘involved’.

126, The effect of honokiol on proinflammatory cytokines (Figure 3) --  the authors wrote that “IFN-γ is the main inducer of IDO activation.  To evaluate whether any activation of IDO in the hippocampal. . .”.  There is no IDO activity data in this manuscript.  The authors should add an activation study in order to claim this association between cytokine profiles and IDO activity.  In addition, in order to claim IFN-γ mediated IDO activation, authors should also show reversal of IDO activity with an IFN inhibitor.

Figures 4, 5, and 6 should be combined.  Immunoblots shown in Figure 5 are not representative of the group data shown in Figure 6.  IDO bands in the model group and honokiol groups look almost identical. NF-kB p65 looks almost unchanged in all groups, and the GADPH bands are uneven.   I am unsure how the authors found significance with such a small SEM using these micrographs?  The authors should provide the entire blots without cropping as supplemental data.

The authors should add clear information regarding the mechanisms of how IDO or NF-kB p65 are involved in the TRP-KYN pathway.  I assume that LPS-mediated increase of kynurenine and quinolinic acid is associated with IDO upregulation, but the mechanistic explanation is lacking.  It may be a good strategy to provide a diagram with an overview and explanation of the TRP-KYN pathway so readers can more readily appreciate the data shown in Figure 7.

Minor:

-Line 56, Papers published in 2006-2009 are not recent.

-Line 90, 346. Authors measured NF-kB and IDO levels in the hippocampus, but IFN levels were analyzed using non-hippocampal brain tissue. Why?

-Figure 1 and 2. Add video as supplemental data.

-Figure 8-11 should be combined as one.

-Name the model group differently, maybe LPS alone?

-Statistics: define * and #.  Which groups are you comparing?

-#32 reference (Lin et al) is missing a journal name.

-Line 329, What was classified as immobility on the tail suspension test?

-Line, 346, “Some brain tissue samples”? what part of the brain?

Author Response

Dear Editors and Reviewers:

Thank you for your letter and for the reviewers’ comments concerning our manuscript entitled “Antidepressant-like effect and mechanism of action of honokiol on the mouse lipopolysaccharide (LPS) depression model” (ID: molecules-498372). Those comments are all valuable and very helpful for revising and improving our paper, as well as the important guiding significance to our researches. We have studied comments carefully and have made correction which we hope meet with approval. Revised portion are marked in red in the paper. The main corrections in the paper and the responds to the reviewer’s comments are as flowing:

Response to Reviewer 4 Comments:

1、This study is largely based on tryptophan metabolism and kynurenine/tryptophan pathway, however the authors overlooked to provide a detailed explanation of their target mechanisms.   For example, what is the rationale for showing IDO and NF-kB p65 together (Figures 4, 5, and 6)? How are they associated? Based on the discussion (Line 236), I assume that NF-kB p64 acts as a transcription factor to promote IDO? In that case, the authors should show downregulation of IDO with NF-kB p65 inhibition. Also, there are multiple subunits and regulators in the NF-kB pathway. Why did the authors choose to investigate NF-kB p65 in particular? The target mechanisms introduced in the Line 65-72 must be specified, rather than using ambiguous terms such as ‘close relationship’ and ‘involved’.

Response:

First of all, I would like to apologize for the ambiguity caused by our imprecise statement. In order to eliminate ambiguity, I made the following changes:

1) Original manuscript Line 235, “In addition, studies have shown that NF-κB can directly or indirectly promote IDO gene expression” was deleted.

Our research mainly aims to show that honokiol affects the metabolism of tryptophan through anti-inflammatory effects, thereby exerting antidepressant effects. In order to intuitively show that honokiol exerts antidepressant mechanism by influencing tryptophan metabolism through anti-inflammatory effects. So, here we show NF-κB together with IDO.

In addition, in order to make it easier for readers to understand, we have added the following diagram to the text (New manuscript page 9 line 251).

Depression has a relatively complicated pathological mechanism, and neuroinflammation is considered to be an important pathological cause of depression. NF-κB is known to be involved in immune and inflammatory responses by modulating the expression of cytokines. Activation of NF-κB may be a pivotal event in pro-inflammatory signal transduction and inhibition of activation of NF-κB can suppress expression of various genes, such as IL-1β, TNF-α and IFN-γ.

TRP is a synthetic raw material of 5-HT. It can synthesize 5-HT under the action of tryptophan hydroxylase and it also can be metabolized to produce active products such as KYN under the action of IDO.

IDO-mediated abnormalities in TRY-KYN metabolism may be associated with the onset of depression. The effects of abnormal metabolism of TRY-KYN mediated by IDO on the pathogenesis of depression include: On the one hand, inflammatory factors cause TRP depletion by activating IDO, resulting in a deficiency of 5-HT, which in turn affects its function. On the other hand, by activating IDO, it promotes the metabolic disorder of tryptophan-kynurenine (TRY-KYN), increases neurotoxic products, activates the glutamatergic system, and increases neurotoxicity. Therefore, we believe that inflammatory factors cause IDO activation is a key link in depression.

2) Due to the critical position of NF-kB p65 in the NF-kB pathway, and the NF-kB p65 detection technology in the related studies is relatively mature, we chose to detect NF-kB p65.

NF-κB is closely associated with neuroinflammation: upon the initiation of inflammatory or oxidative damage, p65 subunit of NF-κB was exposed, which brought about the phosphorylation and translocation of NF-κB to the nucleus subsequently.

NF-κB was originally a nuclear protein that can specifically bind to the enhancer κB sequence of the immunoglobulin kappa light chain gene, which is called NF-κB.To date, at least five NF-κB family members have been identified, namely c-Rel, NF-κB1 (p50), NF-κB2 (p52), RelA (p65) and RelB. A common form of activation is a heterodimer consisting of the p65 and p50 protein subunits of the polypeptide chain. Other activation forms include the p50 homodimer, p65 homodimer. The p50/p65 heterodimer is the main physiological role in cells, so it is customary to refer to the p50/p65 heterodimer as NF-κB. In resting cells, NF-κB is complexed with its inhibitor IκB, and retained in the cytoplasm. When the body is stimulated by external factors such as inflammation or oxidative damage, IκB is phosphorylated by protein kinase and protein phosphatase, and dissociates from NF-κB dimer, exposing the nuclear localization symbol of p50 protein. NF-κB is activated and translocated into the nucleus to bind to specific sites on the DNA strand to initiate gene transcription.

As shown in the figure, due to the critical position of NF-kB p65 in the NF-kB pathway, and the NF-kB p65 detection technology in the related studies is relatively mature, we chose to detect NF-kB p65. In the future experiments, we should indeed consider other subunits and regulators in the NF-kB pathway, and thank you again for your fruitful suggestions.

3) Original manuscript Line 65-72, the statements of “In addition, other studies have shown that there is a close relationship between the tryptophan-kynurenine (TRP-KYN) pathway and depression in the inflammatory state, while indoleamine 2,3-dioxygenase (IDO), an enzyme involved in tryptophan metabolism, is a key factor connecting inflammation and depression. Clinical studies have found that patients with depression have a higher KYN/TRP ratio than non-depressed patients, suggesting that IDO activation is associated with depression. These studies all indicate that neuroinflammation is involved in the development of depression, contributing to a new understanding of the illness.” were corrected as “In addition, other studies have shown that depression in the inflammatory state is often accompanied by a disorder of the tryptophan-kynurenine (TRP-KYN) metabolic pathway, while indoleamine 2,3-dioxygenase (IDO), an enzyme involved in tryptophan metabolism, is a key factor connecting inflammation and depression. Clinical studies have found that patients with depression have a higher KYN/TRP ratio than non-depressed patients, suggesting that IDO activation is associated with depression. These studies all indicate that neuroinflammation is involved in the development of depression, contributing to a new understanding of the illness.” (New manuscript page 2 line 55-62).

2、126, The effect of honokiol on proinflammatory cytokines (Figure 3) --  the authors wrote that “IFN-γ is the main inducer of IDO activation.  To evaluate whether any activation of IDO in the hippocampal. . .”.  There is no IDO activity data in this manuscript.  The authors should add an activation study in order to claim this association between cytokine profiles and IDO activity.  In addition, in order to claim IFN-γ mediated IDO activation, authors should also show reversal of IDO activity with an IFN inhibitor.

Response:

First of all, thank you for your fruitful comments.

We tested IFN-γ levels in the cerebral cortex here to verify whether peripheral administration of LPS causes an increase in IFN-γ levels in the brain. IFN-γ is a potent activator of IDO, and studies have shown that BCG-induced depressive behavior in mice is associated with IFN-γ [1-2]. In addition, our previous studies also showed that IFN-γ can induce depression-like behavior in rats, activate IDO and affect tryptophan metabolism [3]. Therefore, in order to verify whether peripheral administration of LPS causes central IFN-γ release, we detected IFN-γ in the cerebral cortex.

In addition, I want to explain that because there are more detection indicators. Moreover, the mass of hippocampus in mice is small, and the content of tryptophan metabolites is relatively difficult to detect, so we detected the tryptophan metabolites in peripheral blood to reflect central tryptophan metabolism.

In the next experiment, we will further optimize the experiment to make the results more reliable.

1.       Connor, J. C.; André, C.; Wang, Y.; Lawson, M. A.; Szegedi, S. S.; Lestage, J.; Castanon, N.; Kelley, K. W.; Dantzer, R., Interferon-γ and Tumor Necrosis Factor-α Mediate the Upregulation of Indoleamine 2,3-Dioxygenase and the Induction of Depressive-Like Behavior in Mice in Response to Bacillus Calmette-Guérin. The Journal of Neuroscience 2009, 29, (13), 4200.

2.       Wirleitner, B.; Neurauter, G. K.; Frick, B.; Fuchs, D. J. C. M. C., Interferon-gamma-induced conversion of tryptophan: immunologic and neuropsychiatric aspects. Current Medicinal Chemistry 2003, 10, (16), -.

3.       Zhou, J. Y., Li, L. N., Lu, Yi., Fan, A. R., Yu, X., Zhang, J., Li, H. N., Chang, H.S. Antidepressant effects of Peiyuan Jieyu Formula and its regulatory mechanism on TRYKYN metabolic pathway in the rat of depression induced by IFN γ. World Journal of Integrated Traditional and Western Medicine 2016, 11, (08), 1083-1086.

3、Figures 4, 5, and 6 should be combined.  Immunoblots shown in Figure 5 are not representative of the group data shown in Figure 6.  IDO bands in the model group and honokiol groups look almost identical. NF-kB p65 looks almost unchanged in all groups, and the GADPH bands are uneven.   I am unsure how the authors found significance with such a small SEM using these micrographs?  The authors should provide the entire blots without cropping as supplemental data.

Response:

Thank you for pointing out the shortcomings in the article. In order to make the results of the article more convincing, we re-analyzed and typeset (New manuscript page 5 line 143).

4、The authors should add clear information regarding the mechanisms of how IDO or NF-kB p65 are involved in the TRP-KYN pathway.  I assume that LPS-mediated increase of kynurenine and quinolinic acid is associated with IDO upregulation, but the mechanistic explanation is lacking.  It may be a good strategy to provide a diagram with an overview and explanation of the TRP-KYN pathway so readers can more readily appreciate the data shown in Figure 7.

Response:

Thank you very much for your fruitful comments. Based on your suggestions, in order to make it easier for readers to understand, we have added the following diagram to the text (New manuscript page 9 line 250-264).

Figure 7. IDO pathway and tryptophan metabolism pathway.

Notes: Tryptophan catabolism through the kynurenine pathway involves IDO an intracellular rate limiting enzyme. Proinflammatory cytokine is upregulated by LPS regulated by NF-κB. Activation of NF-κB results in the release of pro-inflammatory cytokines, which in turn activates IDO activity. Tryptophan may be metabolized to serotonin or alternatively is metabolized via the kynurenine pathway. Tryptophan is metabolized to kynurenine  by indoleamine 2,3-dioxygenase (IDO) . Kynurenine is then converted via kynurenine aminotransferases (KAT) to kynurenic acid, a neuroprotective molecule as it antagonizes glutamate receptor-induced neurotoxicity. In addition, kynurenine can also be converted to 3-hydroxykynurenine by kynurenine-3-monooxygenase(KMO), for which evidence is accumulating of its neurotoxic capability.  Sequential conversion to 2-amino-3-carboxymuconate-semialdehyde is the penultimate step leading to enzymatic production of (neuroprotective) picolinic acid, and the (non-enzymatic) production of the well-known neurotoxic compound quinolinic acid (QUIN). Further conversion of QUIN to the essential cofactor NAD+ is catalysed by quinolinate phosphoribosyltransferase (QPRT).

5、Minor:

1)-Line 56, Papers published in 2006-2009 are not recent.

Response:

First of all, I would like to apologize for the inaccuracy of my wording.

Original manuscript Line 56, the statements of “Recent” was corrected as “Many” (New manuscript page 2 line 46).

2)-Line 90, 346. Authors measured NF-kB and IDO levels in the hippocampus, but IFN levels were analyzed using non-hippocampal brain tissue. Why?

Response:

We apologize for the misunderstanding caused by our unclear statement.

We tested IFN-γ levels in the cerebral cortex here to verify whether peripheral administration of LPS causes an increase in IFN-γ levels in the brain. IFN-γ is a potent activator of IDO, in order to verify whether peripheral administration of LPS causes central IFN-γ release, we detected IFN-γ in the cerebral cortex.

In addition, the mice hippocampus is of lower quality. In this experiment, the hippocampus of mice was about 20 mg.

Therefore, we separated the cerebral cortex from other brain tissues in order to better detect related indicators.

3)-Figure 1 and 2. Add video as supplemental data.

Response:

In the newly uploaded manuscript, I have compressed and uploaded the video data. The video file is too large, can not be uploaded to the submission system, please find it at the following link: https://zenodo.org/record/2701965#.XNVjLpyd2M8.

4)-Figure 8-11 should be combined as one.

Response:

According to your comments, I made the following changes:

Figure 7. The effect of honokiol on free Ca2+ levels in the brain cells of mice. Data are expressed as mean ± SEM (n = 4). **P < 0.01 LPS+ Honokiol vs LPS.

5)-Name the model group differently, maybe LPS alone?

Response:

Thank you very much for your fruitful comments. According to your comments, I made the following changes:

Original manuscript all the statements of “the model group” was corrected as “ LPS group”; the statements of “the Honokiol group” was corrected as “ LPS+Honokiol group”.

6)-Statistics: define * and #.  Which groups are you comparing?

Response:

I apologize for the misunderstanding you have caused. In order to eliminate misunderstanding, I made the following corrections:

# LPS vs control; *** LPS+ Honokiol vs LPS.

7)-#32 reference (Lin et al) is missing a journal name.

Response:

We are very sorry for our incorrect writing.

All the places without the names of the journals have been corrected and marked red in the references. You can view it in the newly uploaded manuscript.

8)-Line 329, What was classified as immobility on the tail suspension test?

Response:

Mice were considered immobile only when they hung passively and completely motionless.

We apologize for the misunderstanding caused by our unclear statement. According to your suggestion, we have added the following contents, hoping to deepen reader's understanding.

In addition, for a more intuitive display, I specifically selected a map on MedLab-U/4C501H to further deepen your understanding. As shown in the figure, the part shown as a straight line is considered to be the immobility time of the mice in the tail suspension test.

9)-Line, 346, “Some brain tissue samples”? what part of the brain?

Response:

We apologize for the misunderstanding caused by our negligence.

We have made correction according to you comments.

After 4.5.1, mice were killed and their brains were rapidly removed, and the hippocampus and cerebral cortex were isolated. The cerebral cortex was placed in a centrifuge tube and 4°C saline was added at 4 times the volume of tissue.

As you know, the mice hippocampus is of lower quality. In this experiment, the hippocampus of mice was about 20 mg.

Therefore, we separated the hippocampus from other brain tissues in order to better detect related indicators.

Thank you again for your fruitful comments.

We tried our best to improve the manuscript and made some changes in the manuscript. These changes will not influence the content and framework of the paper. And here we did not list the changes but marked in yellow in revised paper.

We appreciate for your warm work earnestly, and hope that the correction will meet with approval.

Once again, thank you very much for your comments and suggestions.

Sincerely,

Bo Zhang

Round  2

Reviewer 1 Report

Authors must clearly explain a choice of honakiol dose for current research, and clearly state doses in Figures along the manuscript.

Author Response

Dear Editors and Reviewers:

Thank you for your letter and for the reviewers’ comments concerning our manuscript entitled “Antidepressant-like effect and mechanism of action of honokiol on the mouse lipopolysaccharide (LPS) depression model” (ID: molecules-498372). Those comments are all valuable and very helpful for revising and improving our paper, as well as the important guiding significance to our researches. We have studied comments carefully and have made correction which we hope meet with approval. Revised portion are marked in yellow in the paper. The main corrections in the paper and the responds to the reviewer’s comments are as flowing:

Response to Reviewer 1 Comments:

Authors must clearly explain a choice of honakiol dose for current research, and clearly state doses in Figures along the manuscript.

Response:

First of all, thank you very much for your fruitful comments. According to your suggestion, I added the following to the manuscript.

Studies have shown that pretreatment with honokiol (10 mg/kg) significantly improved depression-like behaviour induced by chronic restraint stress [75]. In addition, in our previous study [35], we studied the antidepressant-like effects of honokiol on acute and chronic stress mice by setting up three dose groups (2.5, 5, 10 mg/kg). Our results also showed that pretreatment with honokiol (10 mg/kg) significantly improved depression-like behavior ( New manscriptLine 280-284).

Based on our previous experiments, honokiol was orally administered once daily for 11 days at a dose of 10 mg/kg bw.

In addition, according to your request, I clearly indicated the dose of honokiol in the manuscript with figures. Original manuscript the statements of “LPS+Honokiol”were corrected as “LPS+Honokiol (10mg/kg)”.

Thank you again for your fruitful comments.

We tried our best to improve the manuscript and made some changes in the manuscript. These changes will not influence the content and framework of the paper. And here we did not list the changes but marked in yellow in revised paper.

We appreciate for your warm work earnestly, and hope that the correction will meet with approval.

Once again, thank you very much for your comments and suggestions.

Sincerely,

Bo Zhang

Reviewer 4 Report

I really appreciate the improvement in this manuscript.

Author Response

Dear Editors and Reviewers:

Thank you for your letter and for the reviewers’ comments concerning our manuscript entitled “Antidepressant-like effect and mechanism of action of honokiol on the mouse lipopolysaccharide (LPS) depression model” (ID: molecules-498372). Those comments are all valuable and very helpful for revising and improving our paper, as well as the important guiding significance to our researches. We have studied comments carefully and have made correction which we hope meet with approval. The main corrections in the paper and the responds to the reviewer’s comments are as flowing:

Response to Reviewer 4 Comments:

English language and style are fine/minor spell check required.

Response:

    Based on your comments, we have revised the manuscript with reference to the editor's recommended language retouching service (attached: English editing certificate). English spelling and grammar mistakes have been corrected.

    Thank you again for your fruitful comments.

We tried our best to improve the manuscript and made some changes in the manuscript. These changes will not influence the content and framework of the paper.

We appreciate for your warm work earnestly, and hope that the correction will meet with approval.

Once again, thank you very much for your comments and suggestions.

Sincerely,

Bo Zhang
